# A Delphi study to explore and gain consensus regarding the most important barriers and facilitators affecting physiotherapist and pharmacist non-medical prescribing

Emma Graham-Clarke[1]*, Alison Rushton[2,3], John Marriott[1]

1 School of Pharmacy, Institute of Clinical Sciences, College of Medical and Dental Sciences, University of Birmingham, Birmingham, United Kingdom, 2 School of Physical Therapy, Western University, London, Canada, 3 Centre of Precision Rehabilitation for Spinal Pain, School of Sport, Exercise and Rehabilitation Sciences, College of Life and Environmental Sciences, University of Birmingham, Birmingham, United Kingdom

* EMG315@bham.ac.uk

**Data Availability Statement:** All relevant data are within the paper and its Supporting Information files.

## Abstract

Non-medical prescribing was introduced into the United Kingdom to improve patient care, but early research indicated a third of Allied Health Professionals may not use their prescribing qualification. A previous literature review, highlighting factors influencing prescribing, identified only papers with nursing and pharmacy participants. This investigation explored consensus on factors affecting physiotherapist and pharmacist non-medical prescribers. A three round Delphi study was conducted with pharmacist and physiotherapist prescribers. Round One comprised information gathering on facilitators and barriers to prescribing participants had experienced, and underwent content analysis. This was followed by two sequential consensus seeking rounds with participants asked to rate the importance of statements to themselves. Consensus criteria were determined a priori, including median, interquartile range, percentage agreement and Kendall's Coefficient of Concordance (W). Statements reaching consensus were ranked for importance in Round Three and analysed to produce top ten ranks for all participants and for each professional group. Participants, recruited October 2018, comprised 24 pharmacists and 18 physiotherapists. In Round One, content analysis of 172 statements regarding prescribing influences revealed 24 themes. 127 statements were included in Round Two for importance rating (barriers = 68, facilitators = 59). After Round Two, 29 statements reached consensus (barriers = 1, facilitators = 28), with no further statements reaching consensus following Round Three. The highest ranked statement in Round Three overall was: "Being able to prescribe to patients is more effective and really useful working [in my area]". Medical support and improved patient care factors appeared the most important. Differences were noted between physiotherapist and pharmacist prescribers regarding the top ten ranked statements, for example team working which pharmacists ranked higher than physiotherapists. Differences may be explained by the variety of practice areas and relative newness of physiotherapy prescribing. Barriers appear to be post or person specific, whereas facilitators appear universal.

**Funding:** The author(s) received no specific funding for this work.

**Competing interests:** The authors have declared that no competing interests exist.

## Introduction

Non-medical prescribing (NMP) (prescribing by professions other than the medical profession) was introduced in the United Kingdom (UK) to improve patient care and access to medicines, following the second Crown report [1]. The UK recognises two main approaches to NMP; supplementary and independent. Supplementary prescribers can only prescribe from a clinical management plan agreed by the doctor treating the patient, supplementary prescriber and patient [2]. Independent prescribers are responsible for patient care, including assessment and prescribing [3] and may prescribe any drugs detailed by profession specific legislation and regulations [4]. Initially only nurses and pharmacists could become non-medical prescribers, gaining independent prescribing rights in 2006. Subsequently there has been a gradual expansion to other professions [5, 6].

Since NMP introduction, with the UK National Health Service (NHS) experiencing increased patient demand, workforce shortage pressures and funding shortfalls, the policy emphasis has changed to streamlining care [5, 7, 8]. For example, physiotherapists are moving into first point of contact roles for patients with musculoskeletal problems, where the ability to prescribe enables them to provide a complete treatment package without referral to other healthcare professionals [8–10]. These plans will be hindered if non-medical prescribers are deterred from utilising their skills. Additionally, the approximate cost of training non-medical prescribers was calculated as £10,000; failure to utilise this skill therefore represents poor use of limited NHS funds [11].

Previous research evaluating the use of NMP indicated that approximately a third of qualified Allied Health Professional prescribers may not prescribe compared to approximately 10% of nurses [11, 12]. A systematic literature review described 15 factors or themes (for example, medical support or facilities availability) potentially influencing prescribing utilisation by non-medical professions [13]. The majority of included studies concerned nurse prescribing and the remainder pharmacists. No papers reviewed the experiences of other non-medical prescribers; hence it is unclear if other NMP professions experience similar factors affecting prescribing utilisation. Establishing factors that facilitate or prevent NMP and investigating if these are generic to the different NMP professions, or are professional, situational or person specific will aid NMP development.

This paper presents the results of an investigation into facilitators and barriers encountered by two NMP professions, pharmacy and physiotherapy. These professions were chosen as they are similar sizes in the UK (approximately 50,000), may work individually or as teams, and may work in all healthcare sectors [14, 15]. They differ in the length of time that each profession had prescribing rights, with pharmacy gaining independent prescribing rights six years earlier than physiotherapy [16, 17].

The primary objective was to gain consensus regarding the factors that have supported, or discouraged, pharmacist and physiotherapist non-medical prescribers from utilising their prescribing qualification. Furthermore, to determine which factors had greatest influence on prescribing utilisation, and if these factors were perceived similarly between pharmacists and physiotherapists.

## Method

### Design

Research methods, such as consensus techniques, that systematically obtain and prioritise expert opinion can be utilised when published information is scanty or non-existent [18, 19]. The Delphi technique was developed in the 1950s as a forecast method and has been

increasingly used in healthcare research [20]. It is an iterative technique using sequential questionnaires and controlled group feedback, with anonymity of participants to each other as a key feature [21, 22]. The classic Delphi design has an information seeking first-round followed by prioritisation rounds, stopping when consensus is achieved. The literature describes variations, such as using literature reviews to generate the first round [20]. A previous systematic literature review [13], showing an absence of physiotherapist literature, indicated the appropriateness of the classic Delphi information gathering first round to seek physiotherapy opinions [22].

Questionnaires were administered using online survey software (https://www.onlinesurveys.ac.uk/) supporting participant anonymity whilst providing response tracking and automatic reminder facilities. The study was approved by the University of Birmingham's Science, Technology, Engineering and Mathematics Ethical Review Committee and all data were held securely in accordance with university guidance. The study is reported in accordance with the criteria proposed by Jünger and colleagues, in the absence of an agreed reporting structure for Delphi studies (S1 Appendix) [23].

## Participants

Delphi participants are described as 'experts' and require knowledge of the research topic. A criterion based purposive technique was adopted to recruit pharmacist and physiotherapist independent prescribers, qualified since 2013 when the law was amended to permit physiotherapist independent prescribing, using a sample matrix (S1 Table) [17, 24, 25]. Readily accessible lists of such prescribers are unavailable, and recruitment was conducted indirectly. Invitation emails were sent to West Midlands NMP Leads, CHAIN (a healthcare orientated online mutual support network: www.chain-network.org.uk) and Health Education England (a national body overseeing education: https://www.hee.nhs.uk) Pharmacy Deans, requesting they forward the email invitation to physiotherapist and pharmacist prescribers. Invitations to participate contained a brief study outline, participant information sheet and contact details. Potential participants were invited to contact the lead researcher with questions and to express their interest in participation. Sample sizes for Delphi exercises are variable, ranging from fewer than 10 to several hundred, with smaller numbers suitable for homogenous samples [21]. The current research sample was heterogenous since recruitment covered all healthcare sectors and levels of experience. As the number of qualified physiotherapist independent prescribers was unknown, a pragmatic target sample size of 30 for each profession was chosen. Recruitment was closed in October 2018.

## Procedure and analysis

A three round Delphi was conducted, following the scheme in Fig 1. People responding positively to the invitation email were sent an email link to the first questionnaire. Subsequent questionnaires were sent to participants who responded to the previous questionnaire. Each round was open for one month, with non-respondents sent reminder emails at two and three weeks to maximise response rate [26–29]. Regular emails regarding the progress of the exercise were sent to all participants to minimise response dropout; an acknowledged limitation of Delphi studies [27, 28]. The Round One questionnaire was piloted with nurse independent prescribers and the questionnaires for Rounds Two and Three were reviewed by the research group.

**Round One.** The Round One questionnaire comprised three sections (see S2 Appendix). The first section included study information and a consent statement; participants could only proceed further if consent was agreed. The second section requested brief demographic data.

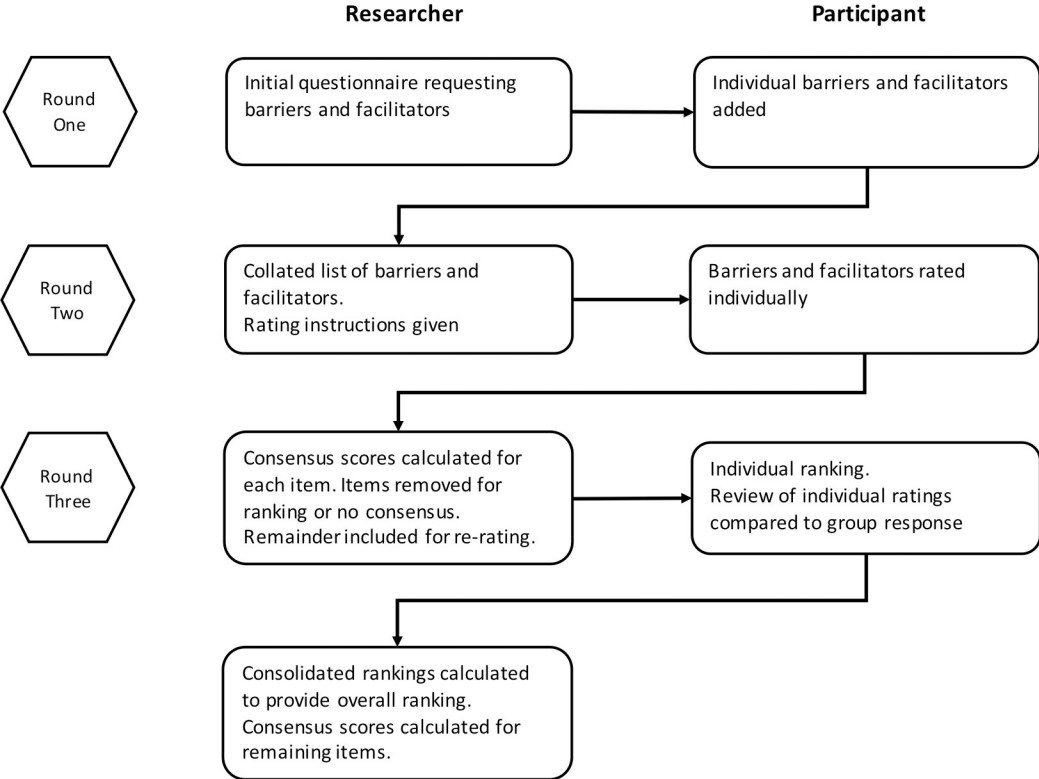

**Fig 1. Diagram describing the three Delphi rounds and the researcher and participant actions at each round.**

The third section, using open ended questions, asked participants to provide at least three facilitators and/or barriers to prescribing that they had encountered. Participants were able to comment on questionnaire design and content.

Demographic data were imported into SPSS (IBM® SPSS® Statistics 25) for descriptive statistics. The open responses, detailing barriers and facilitators, were exported into NVivo® 12 (QSR International) for content analysis [30–32]. The responses were reviewed and coded to identify recurrent themes and used to develop the Round Two questionnaire [21, 26, 33].

**Round Two.** The Round Two questionnaire comprising the tabulated statements was sent to all participants who had responded to Round One (see S3 Appendix). Participants were asked to rate the importance of the factors in each statement to their practice through a 5-point Likert scale. [28, 34–36] and were able to add free text comments throughout to explain their ratings.

Anonymous rating data were exported from the online survey software into an Excel spreadsheet (Microsoft® Excel for Mac 16) and thence into SPSS (IBM® SPSS® Statistics 25). Percentage agreement, median and interquartile range (IQR) were calculated for each statement [21, 22, 28, 35]. The median and IQR were chosen as they are appropriate for ordinal scales such as Likert [18, 21, 22, 28]. Kendall's Coefficient of Concordance (W) was calculated as a measure of group response agreement [22, 37, 38]. Kendall's Coefficient of Concordance (W) results range from 0 (no agreement) to 1 (full agreement). Consensus criteria, based on previous studies, were determined a priori (Table 1) [28, 35, 36, 39].

**Round Three.** The Round Three questionnaire was derived following analysis of Round Two, using the decision criteria listed in Table 2. and was sent to all participants who had completed Round Two (see S4 Appendix). Participants received group median feedback on

**Table 1. Consensus criteria for Rounds Two and Three.**

| Test | Round Two | Round Three |
|---|---|---|
| Percentage agreement | 60 | 70 |
| Median | 3.5 | 4 |
| Interquartile range | ≤2 | ≤1 |
| Kendall's Coefficient of Concordance (W) | P<0.05 | P<0.05 |

statements included for re-rating and were invited to review and amend their rating, using the same 0–5 Likert scale. Statements achieving consensus in Round Two were included separately, with participants asked to rank the ten most important to them, from one to ten.

Consensus criteria analyses were calculated as described in Round Two. The number of comments received in Round Two and Round Three were compared, with a decrease in numbers supporting stability in participant responses [40]. The ranking data were exported into an Excel spreadsheet (Microsoft® Excel for Mac 16) and weighted sum ranks calculated, allowing ordering of statements (See S5 Appendix).

## Results

### Demographic data

Forty-nine participants expressed an interest in participating and received the Round One questionnaire. The Round One questionnaire was completed by 42 participants (n = 24 pharmacists, n = 18 physiotherapists). Participant demographic data is presented in Table 3. The majority of physiotherapists (11/18) had been qualified in their profession for ≥ 21 years, compared to pharmacists (6/24). Secondary care was the predominant practice area for recruited pharmacists (21/24), with physiotherapist practice areas distributed across all sectors. Physiotherapists were also more likely to have a secondary practice area (7/18) than pharmacists (1/24). More pharmacists were active prescribers (20/24) compared to physiotherapists (11/18).

### Round One results

The number of statements received from each participant ranged between three and seven, with 172 in total. Content analysis resulted in 24 major themes (see Table 4). Following removal of duplicates, 127 statements were included in Round Two across the 24 themes (59 facilitators, 68 barriers). In many cases, participants elaborated on the statement using a free text facility. For example, participant Pharm17 listed '*effective personal development reviews*' as a facilitator and expanded on it as follows:

'*effective PDR enable (sic) to identify areas of development and opportunities for expansion of areas of practice*' Pharm17

**Table 2. Decision criteria regarding statement inclusion in Round Three.**

| Decision | Criteria |
|---|---|
| Included for ranking | Met all consensus criteria, for all participants and for individual professional groups |
| Included for re-rating | Met two consensus criteria and/or disagreement between groups (all participants, individual professional groups) |
| Removed from study | Met one or no consensus criteria, for all participants and for individual professional groups |

**Table 3. Participant demographic data.**

| Demographics | | Pharmacists (n = 24) | | Physiotherapists (n = 18) | | Total (n = 42) | |
|---|---|---|---|---|---|---|---|
| | | n | % | n | % | n | % |
| Years qualified in profession | ≤5 | 2 | 8.3 | 0 | 0 | 2 | 4.8 |
| | 6–10 | 7 | 29.2 | 1 | 5.5 | 8 | 19.0 |
| | 11–15 | 4 | 16.7 | 3 | 16.7 | 7 | 16.7 |
| | 16–20 | 5 | 20.8 | 3 | 16.7 | 8 | 19.0 |
| | >21 | 6 | 25 | 11 | 61.1 | 17 | 40.5 |
| Time qualified as independent prescriber | ≤12 months | 7 | 29.2 | 5 | 27.8 | 12 | 28.6 |
| | >12 months | 17 | 70.8 | 13 | 72.2 | 30 | 71.4 |
| Home nation in which they qualified | England | 23 | 95.8 | 18 | 100 | 41 | 97.6 |
| | Scotland | 1 | 4.2 | 0 | 0 | 1 | 2.4 |
| | Wales | 0 | 0 | 0 | 0 | 0 | 0 |
| | Northern Ireland | 0 | 0 | 0 | 0 | 0 | 0 |
| Main practice area | Primary Care | 3 | 12.5 | 5 | 27.8 | 8 | 19.0 |
| | Secondary care | 21 | 87.5 | 6 | 33.3 | 27 | 64.3 |
| | Community | 0 | 0 | 5 | 27.8 | 5 | 11.9 |
| | Other | 0 | 0 | Private practice 1 Mental health services for older people 1 | 11.1 | 2 | 4.8 |
| Secondary practice areas | Primary Care | 0 | 0 | 1 | 5.5 | 1 | 2.4 |
| | Secondary care | 0 | 0 | 0 | 0 | 0 | 0 |
| | Community | 1 | 4.2 | 4 | 22.2 | 5 | 11.9 |
| | Other | 0 | 0 | Private practice 1 Out-patients 1 | 11.1 | 2 | 4.8 |
| Active prescriber | Yes | 20 | 83.3 | 11 | 61.1 | 31 | 73.8 |
| | No | 4 | 16.7 | 7 | 38.9 | 11 | 26.2 |
| Average number of prescriptions written per week* | <5 | 5 | 20.8 | 7 | 38.9 | 12 | 28.6 |
| | 6–15 | 7 | 29.2 | 3 | 16.7 | 10 | 23.8 |
| | 16–25 | 2 | 8.3 | 1 | 5.5 | 3 | 7.1 |
| | 26–35 | 3 | 12.5 | 0 | 0 | 3 | 7.1 |
| | 36–45 | 1 | 4.2 | 0 | 0 | 1 | 2.4 |
| | >46 | 2 | 8.3 | 0 | 0 | 2 | 4.8 |
| Type of practice§ | Generalist | 10 | 41.7 | 7 | 38.9 | 17 | 40.5 |
| | Specialist | 13 | 54.2 | 11 | 61.1 | 24 | 57.1 |
| Specialities listed | | Anticoagulation | Critical care and respiratory | | | | |
| | | Antimicrobials | MSK and pain | | | | |
| | | Clinical research/ cardiology | Pain Management (n = 2) | | | | |
| | | Critical care | Pain management and community acquired infections | | | | |
| | | Diabetes and Hypertension | Persistent pain | | | | |
| | | Heart Failure | Respiratory | | | | |
| | | Infections | Rheumatology | | | | |
| | | Mental Health | Spinal orthopaedic services | | | | |
| | | Nephrology | | | | | |
| | | Neuro-developmental disorders | Stroke | | | | |
| | | Osteoporosis | Stroke/Neurology | | | | |
| | | Palliative care | | | | | |
| | | Respiratory Medicine | | | | | |

**Table 4. Identified themes following content analysis of Round One results.**

| Theme | Description | Facilitator (n) | Barrier (n) |
|---|---|---|---|
| **Alternative prescriber** | As alternative to a doctor, or replaced by an alternative, possibly 'cheaper' model | 2 | 3 |
| **Clinical skills** | Clinical examination skills–acquisition or lack of. | 1 | 2 |
| **Confidence** | Personal confidence in skills | 2 | 2 |
| **Employer** | Support from Trust, department, manager etc | 12 | 5 |
| **Funding** | Funding to practice | 0 | 5 |
| **Information sources** | Access to information sources, use of information sources. Keeping up to date with new information. | 3 | 2 |
| **Infrastructure** | Access to clinic room, prescription pads etc. | 2 | 2 |
| **Knowledge** | Experience in prescribing area (or lack of). Specialist knowledge. | 6 | 1 |
| **Legal Aspects** | Prescribing legislation, indemnity, registration | 4 | 9 |
| **Medical Records** | Access to medical records—paper or electronic | 3 | 5 |
| **Medical support** | Medical support—GP/Consultant etc. Includes acceptance of role etc.. | 19 | 6 |
| **Nursing support** | Relationship with nursing staff. Could be supportive or indicate lack of understanding of the role. | 2 | 2 |
| **Patients** | Patient experience and knowledge of NMP. | 5 | 0 |
| **Peer support** | Other colleagues and clinicians. | 12 | 5 |
| **Post Course Support** | Post course development including appraisals | 3 | 2 |
| **Prescribing budget** | Access to prescribing budget | 1 | 1 |
| **Prescribing Course** | Usefulness/appropriateness of course. Aspects relating to communication from the university during and following course completion. | 0 | 3 |
| **Prescription review** | Pharmacy review of prescriptions. Includes need for second pharmacist. | 1 | 5 |
| **Role** | Personal job role. Includes effect of change in role. | 2 | 7 |
| **Role model** | Acting as a role model. Being inspired by other role models. | 2 | 0 |
| **Time** | Time to prescribe, time free from other duties etc. | 0 | 10 |
| **Ward round** | Role and attendance on ward rounds. Attendance at MDT meeting. | 1 | 2 |
| **Working environment** | Totality of working environment, including protocols and policies guiding activity. | 2 | 3 |
| **Minor themes** | Competency, formulary, practice area, external drivers and working patterns | 1 | 4 |

Likewise, Physio05 gave '*the Law*' as a barrier, elaborating with:

'*as a physio I am restricted to my prescribing. In most terms this is appropriate but it does cause me to have to go to a GP for a prescription that I may have been able to do myself*' Physio05

## Round Two results

Of participants completing Round One, n = 31 responded in Round Two. Kendall's W was calculated with the significance results indicating agreement between participants as a whole and for pharmacists and physiotherapists separately (Table 5).

Twenty-nine statements reached consensus and included 28 facilitator and one barrier statement. Of the 40 statements not reaching the consensus criteria, 10 were facilitators and 30 barriers and were removed from further rounds as described in Table 2. The remaining statements were included for re-rating in Round Three. Full results are presented in supporting information S2 and S3 Tables. Comments were received for most statements, with 300 received for facilitators (range 0–16 per statement), and 134 received for barriers (range 0–6 per statement). Comments included requests for more explanation (5% of all comments) or indicated that the statement was irrelevant to themselves or their practice (facilitator statements—30%, barrier statements—43%).

**Table 5. Kendall's Coefficient of Concordance (W) results for Round Two.**

| Group | Population | n | Kendall's W | Chi-Square | df | Significance |
|---|---|---|---|---|---|---|
| All statements | Total group | 31 | .284 | 1110.893 | 126 | <0.01 |
| | Pharmacists | 14 | .393 | 692.609 | 126 | <0.01 |
| | Physiotherapists | 17 | .294 | 629.334 | 126 | <0.01 |
| Facilitator statements | Total group, | 31 | .234 | 420.712 | 58 | <0.01 |
| | Pharmacists, | 14 | .333 | 270.610 | 58 | <0.01 |
| | Physiotherapists | 17 | .230 | 226.642 | 58 | <0.01 |
| Barrier statements | Total group | 31 | .090 | 187.220 | 67 | <0.01 |
| | Pharmacists | 14 | .223 | 209.178 | 67 | <0.01 |
| | Physiotherapists | 17 | .151 | 171.609 | 67 | <0.01 |

## Round Three results

Of the 31 participants receiving the Round Three questionnaire, 20 responded. No further statements reached consensus following re-rating (see S4 and S5 Tables). Round Three Kendall's W is reported in Table 6, indicating agreement except for the facilitator statements from physiotherapists. Fewer comments were received, compared with Round Two, indicating stability within responses (30 for facilitators [range 0–4 per statement], 11 for barriers [range 0–1 per statement]). However, a small number of comments indicated a failure to understand the limitations imposed on selected professions. For example, a pharmacist responded to the statement: "Lack of medical cover at times means I cannot prescribe opioids" with:

*"Why would this be an issue?"* Pharm12

Table 7 reports Kendall's W for the ranking exercise and indicates agreement within groups (p>0.05). Table 8 lists the weighted rank sums, for all participants and each profession. The ranks for all participants are presented graphically in Fig 2 and for each profession in Fig 3. The highest ranked statement was common to all participants and to each profession:

"***Being able to prescribe to patients is more effective and really useful working [in my area]***"

Differences are noted when the top ten ranked statements from all participants are compared with either the pharmacist or physiotherapist groups. Statements made by the

**Table 6. Kendall's Coefficient of Concordance (W) results for Round Three re-rating of statements.**

| Group | Population | n | Kendall's W | Chi-Square | df | Significance |
|---|---|---|---|---|---|---|
| All statements | Total group | 20 | .207 | 236.360 | 57 | <0.01 |
| | Pharmacists | 10 | .302 | 172.251 | 57 | <0.01 |
| | Physiotherapists | 10 | .306 | 174.689 | 57 | <0.01 |
| Facilitator statements | Total group, | 20 | .071 | 28.235 | 20 | .104 |
| | Pharmacists, | 10 | .191 | 38.165 | 20 | .008 |
| | Physiotherapists | 10 | .122 | 24.444 | 20 | .224 |
| Barrier statements | Total group | 20 | .128 | 92.162 | 36 | <0.01 |
| | Pharmacists | 10 | .287 | 103.400 | 36 | <0.01 |
| | Physiotherapists | 10 | .231 | 83.039 | 36 | <0.01 |

**Table 7. Kendall's Coefficient of Concordance (W) for ranked statements.**

| Population | n | Kendall's W | Chi-Square | df | Significance |
|---|---|---|---|---|---|
| Total group | 20 | .132 | 73.812 | 28 | <0.01 |
| Pharmacists | 10 | .185 | 51.761 | 28 | .004 |
| Physiotherapists | 10 | .168 | 47.014 | 28 | .014 |

pharmacist group concur with the top ten statements from all participants, albeit in a different rank order. When the top ten statements for physiotherapists and all participants are compared, three statements differ. In the pharmacist top ten, all weighted sums for statements were ≥30, however only the top five for physiotherapists were ≥30. The weighted sums for remaining statements for physiotherapists were low, with tied ranks affecting 17 statements.

## Discussion

This is the first study to identify the factors influencing the uptake and utilisation of prescribing by physiotherapists and pharmacists and to investigate if each profession perceived them similarly. A similar number of barriers and facilitators were identified in Round One. Following Round Two, consensus was obtained for 28/59 facilitator statements, but only 1/68 barrier statements, with no further consensus achieved after Round Three. It is striking that despite the greater initial number of barrier statements, only one achieved consensus. This suggests that most NMP barriers are specific to the post and person, whereas facilitators are generic.

Of the themes identified from content analysis, 13 had statements achieving consensus. "Medical professionals" was the most highly cited theme, reinforcing the importance of their support for NMP identified in a previous literature review [13]. A disproportionately high number of medical professional statements reached consensus (7/29) in Round Two compared with other themes. Similar numbers of statements relating to patient care (4/29) and employer (4/29) themes achieved consensus. Themes such as medical records and infrastructure highlighted in a previous literature review [13] did not have statements reaching consensus.

The ranking results emphasised the importance of prescribing to patient care, with the foremost statement overall concerning the effectiveness of prescribing for patients. Both professions highlighted the benefit of streamlining care for patients. Additionally, pharmacists ranked highly the statement regarding motivation to help patients benefit from reduced delay and duplication, possibly driven by perceived secondary care hinderances in prescribing medication. Pharmacists and physiotherapists ranked practice related statements in their top ten statement ranking, highlighting the importance to their role. In particular these related to the benefit of a specialist area in allowing the development of skills and knowledge and building confidence. Both professions ranked good working relationships with consultants in their top ten. Subtle differences in the manner in which pharmacists and physiotherapists practice were highlighted by the distribution of statements in the top ten. Pharmacists ranked the three statements mentioning teams in their top ten (direct contact with medical team, working as part of a multidisciplinary team and support from team) showing the importance of team working in their practice. In comparison the physiotherapist top ten highlighted the benefits of multidisciplinary teams but also supportive nursing and medical colleagues, suggesting a more independent mode of working. Only physiotherapists ranked an employer support statement in their top ten statement ranking, which may be driven by the newness of prescribing to physiotherapists and the need for employer support. In comparison, several pharmacists commented that they had changed employer since qualifying as an independent prescriber. Outside the top ten, the weighted rank sums for the remaining statements for both groups were small; rendering

**Table 8. Weighted sums and statements ranks for all participants and each profession.**

| Statements | All participants (n = 20) | | Pharmacists (n = 10) | | Physiotherapists (n = 10) | |
|---|---|---|---|---|---|---|
| | Weighted sum | Rank | Weighted sum | Rank | Weighted sum | Rank |
| Being able to prescribe to patients is more effective and really useful working [in my area] | 917 | 1 | 201 | 1 | 280 | 1 |
| Having a speciality allows development of skills and knowledge | 164 | 2 | 38 | 8 | 94 | 2 |
| Direct contact with medical team caring for patient | 160 | 3 | 95 | 2 | 17 | 14 |
| Motivation to help the patients who will benefit with prescribing and cut care delay / duplication | 157 | 4 | 90 | 3 | 13 | 16 |
| Patient requirements. A need for patients to have streamlined care by being able to prescribe at the point of contact | 139 | 5 | 36 | 9 | 39 | 4 |
| Good relationship with consultants | 128 | 6 | 46 | 5 | 30 | 5 |
| Working as part of an MDT [multidisciplinary team] / interdisciplinary group | 90 | 7 | 40 | 7 | 24 | 6 |
| Personal confidence in specialism | 88 | 8 | 45 | 6 | 21 | 9 |
| Well supported by team and they allow me to prescribe for their patients | 69 | 9 | 60 | 4 | 9 | 20 |
| My knowledge of medication | 62 | 10 | 34 | 10 | 24 | 6 |
| Supportive nursing colleagues | 54 | 11 | 5 | 24 | 43 | 3 |
| Easy access to medication info | 49 | 12 | 19 | 12 | 18 | 12 |
| Clinical supervision with a [doctor] has massively helped me increase my confidence prescribing | 44 | 13 | 14 | 17 | 16 | 15 |
| My employer has provided the support for me to be able to go on the NMP course and then supported me once qualified | 44 | 13 | 16 | 14 | 24 | 6 |
| Forward thinking DMP [designated medical practitioner] who is keen to integrate different MDG [multidisciplinary group] professionals into the team | 38 | 15 | 14 | 17 | 18 | 12 |
| Lack of time to develop further prescribing skills | 35 | 16 | 15 | 15 | 20 | 11 |
| Supportive medical colleagues | 32 | 17 | 3 | 25 | 21 | 9 |
| Great antibiotic guidelines in this trust/area | 27 | 18 | 20 | 11 | 7 | 23 |
| Support from the employer/department for the role of non-medical prescribers | 26 | 19 | 15 | 15 | 11 | 18 |
| Doctors have been working [with] this [NMP] model | 19 | 20 | 9 | 21 | 10 | 19 |
| Management support enables funding and training time to qualify as a prescriber | 19 | 20 | 10 | 20 | 9 | 20 |
| Supportive working environment [with NMP] policies in place | 18 | 22 | 12 | 19 | 6 | 24 |
| Support from my line manager | 17 | 23 | 17 | 13 | 0 | 28 |
| The nature of the role facilitates prescribing practice as part of the overall review of patients | 16 | 24 | 7 | 22 | 9 | 20 |
| Supportive medical supervision / mentorship | 13 | 25 | 0 | 28 | 13 | 16 |
| Wide variety of options that you can offer patients to improve their experience | 13 | 25 | 7 | 22 | 6 | 24 |
| The law enables me to practice as an NMP | 9 | 27 | 1 | 27 | 6 | 24 |
| Support from other NMPs | 4 | 28 | 0 | 28 | 4 | 27 |
| Mentor already NMP—creates a positive environment for NMP | 3 | 29 | 3 | 25 | 0 | 28 |

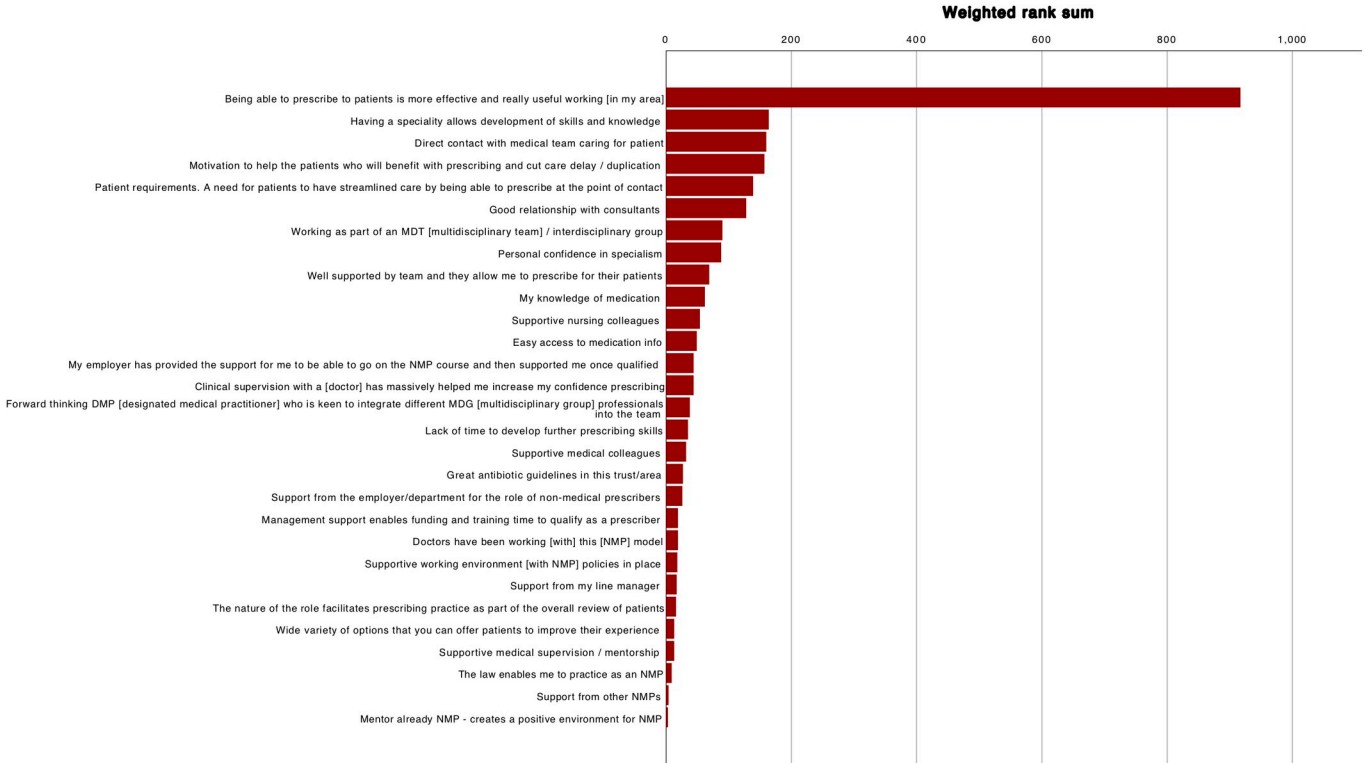

**Fig 2. Ranked statements for all participants by weighted rank sum.**

them inappropriate as discriminators. The only barrier reaching consensus concerned the lack of time to develop skills and was ranked 16th overall and outside the top ten for both pharmacists and physiotherapists, implying that while this was a concern, it was not a major deterrent to prescribing.

Equal numbers of pharmacists and physiotherapists completed Round Three, with potentially the same weighted rank sum, so it is surprising that the physiotherapist weighted rank sums were relatively low compared with pharmacy results. This may be explained by the variety of physiotherapy practice areas and associated factors indicated by the participants. This compares to pharmacists who were primarily recruited from secondary care.

Initially more pharmacist than physiotherapist prescribers were recruited, reflecting both difficulty in accessing physiotherapist prescribers and differences in prescribing legislation dates [17, 41]. Physiotherapists were more likely to have been registered in their profession longer than pharmacists. This reflects previous early prescribing studies which suggested that more experienced professionals adopted prescribing initially after its introduction to their profession [42–46]. Recruited physiotherapists worked in several healthcare settings, whereas pharmacists were mainly from secondary care. Pharmacists were more inclined to be active prescribers, which may reflect how embedded pharmacist prescribing has become, although several comments indicated that pharmacists were now in roles that did not support prescribing.

The relevance of the topic was indicated by the Round One response rate (85%), and the number of barriers and facilitators initially identified. Comments received for each round supported the high engagement level of the participants. Despite steps taken to minimise drop-out, the response rate decreased over the three rounds, with a final response rate representing

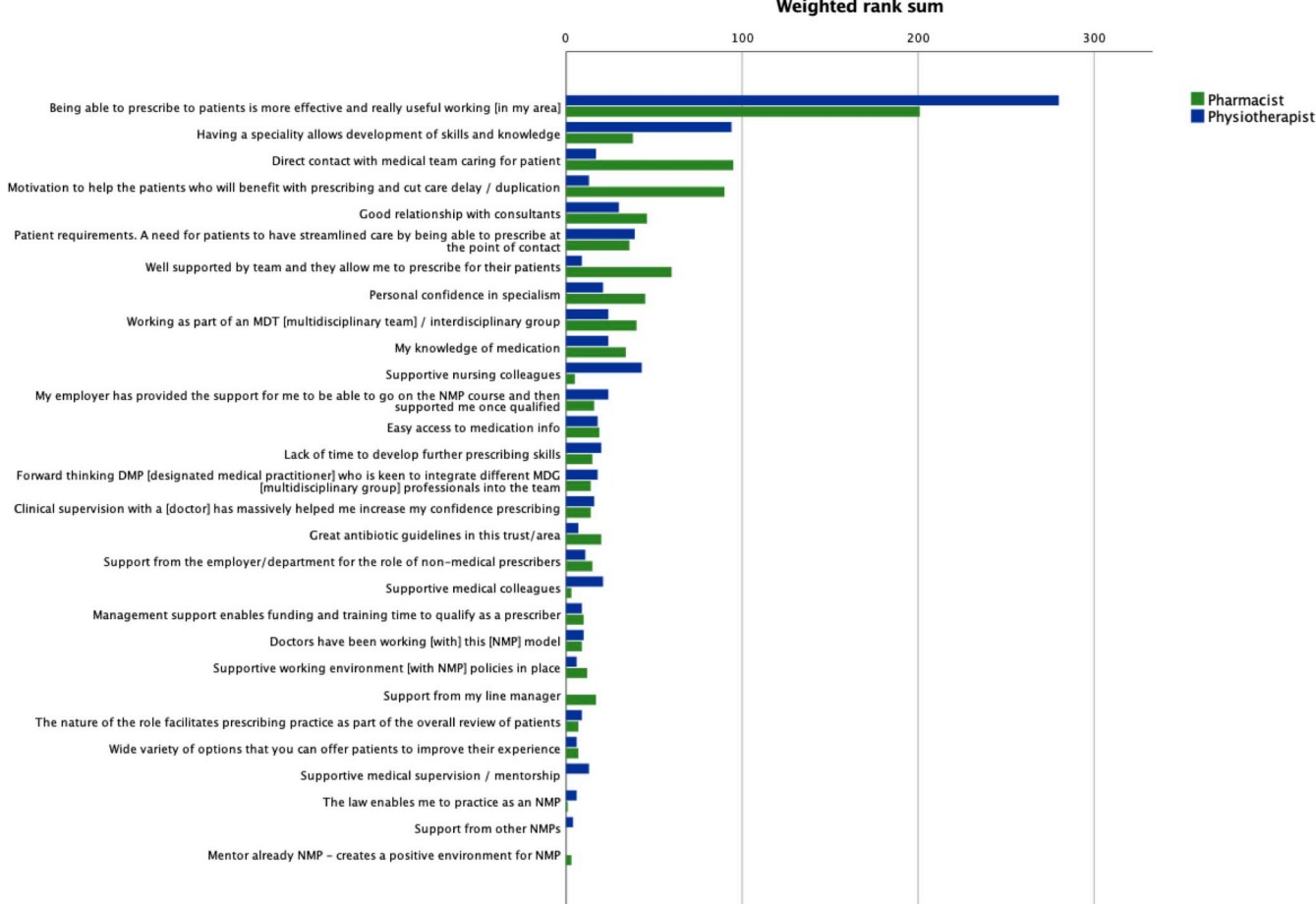

**Fig 3. Ranked statements for professional groups by weighted rank sum.**

41% of the initial 49 participants. The survey software enabled the overall progress through each questionnaire to be reviewed, indicating the potential for questionnaire design and survey software constraints to contribute to the attrition. For Round Two, a balance was required between returning all the statements back to participants, risking disengagement if apparent repetition, and grouping similar statements as a single statement, risking introducing researcher bias [26, 38]. Consequently, the decision was made to only omit those where there was evident duplication. Supported by participant comments in Round Two, statements were removed from Round Three (as described earlier) rendering the questionnaire more manageable, whilst accepting the potential introduction of bias [26, 38]. The survey software constraints resulted in sub-optimal display for the ranking question, with participants commenting that selecting their top ten was challenging.

A small number of comments were received from pharmacist participants indicating they were unaware of prescribing constraints for some professions, or they had forgotten there were physiotherapy participants. Failure to understand these constraints is concerning as it indicates that pharmacists, responsible for dispensing prescriptions, are unfamiliar with prescribing regulations [4].

The two professions were initially selected because of the difference in independent prescribing implementation stage, with pharmacists having a six-year potential advantage over

physiotherapists. This time difference is most apparent when the participant demographics are reviewed, with physiotherapists tending to be both more experienced practitioners and less likely to be actively prescribing compared to pharmacists. However, when the ranked statements are reviewed the differences between the groups would appear to be more related to practice areas and mode of practice, than to prescribing implementation stage. The exception is the support from employers that the physiotherapist group ranked in their top ten, whereas for pharmacists this was not perceived to be as important an issue.

### Strengths and limitations

This is the first study investigating and comparing prescribing barriers and facilitators in pharmacy and physiotherapy professions. The participants' level of engagement, emphasised by the Round One responses and free text comments, highlight the relevance of the topic.

The recruitment strategy relied on self-identifying participants, potentially introducing bias as participants with strong views are more inclined to volunteer [26]. Accessing physiotherapist prescribers also proved difficult, with an initial imbalance in participant numbers. Participant fatigue and attrition are recognised Delphi limitations [27, 28] and this was evident, despite approaches to minimise attrition. Software limitations influenced questionnaire design, deterring participants from completing Round Two and Three, and affecting response rate.

### Conclusion

This study set out to explore the factors (both facilitators and barriers) that affected pharmacist and physiotherapist prescribing, and to determine if there were differences between the two professional groups. Initially similar numbers of facilitator and barrier statements were identified by participants, but only one barrier statement reached consensus, compared to 28 facilitator statements. Improving patient care and medical professionals' support appear to be the most important factors in enabling non-medical prescribing. In contrast the lack of time to develop prescribing skills was the only barrier to reach consensus. These results indicate that prescribing barriers are post and person specific, whereas facilitators are more likely to be generic. Differences in the ranking of facilitator statements were detected between pharmacy and physiotherapy, appearing to reflect the manner in which the two professions practice. In particular pharmacists favoured factors relating to team support whereas these were less important for physiotherapists who may work more independently. This intimates that factors identified in a previous literature review [13] may not be universally applicable to all NMP professions. Participants' opinions shape Delphi results and further research is required to determine the transferability of these results [20, 47].

### Supporting information

**S1 Appendix. Reporting criteria.**
(DOCX)

**S2 Appendix. Round One questionnaire.**
(PDF)

**S3 Appendix. Round Two questionnaire.**
(PDF)

**S4 Appendix. Round Three questionnaire.**
(PDF)

**S5 Appendix. Weighted rank sum example.**
(DOCX)

**S1 Table. Sample matrix for selecting Delphi participants.**
(DOCX)

**S2 Table. Consensus results for facilitator statements, Round Two–grouped by all participants and for each profession.**
(DOCX)

**S3 Table. Consensus results for barrier statements, Round Two–grouped by all participants and for each profession.**
(DOCX)

**S4 Table. Consensus results for facilitator statements, Round Three–grouped by all participants and for each profession.**
(DOCX)

**S5 Table. Consensus results for barrier statements, Round Three–grouped by all participants and for each profession.**
(DOCX)

## Author Contributions

**Conceptualization:** Emma Graham-Clarke, Alison Rushton, John Marriott.

**Data curation:** Emma Graham-Clarke.

**Formal analysis:** Emma Graham-Clarke.

**Investigation:** Emma Graham-Clarke.

**Methodology:** Emma Graham-Clarke, Alison Rushton, John Marriott.

**Project administration:** Emma Graham-Clarke.

**Supervision:** Alison Rushton, John Marriott.

**Validation:** Alison Rushton, John Marriott.

**Visualization:** Emma Graham-Clarke.

**Writing – original draft:** Emma Graham-Clarke.

**Writing – review & editing:** Emma Graham-Clarke, Alison Rushton, John Marriott.

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
