## [Decision Letter · Decision Letter 0]

15 Oct 2020

PONE-D-20-27040

A Delphi study to explore and gain consensus regarding the most important barriers and facilitators affecting physiotherapist and pharmacist non-medical prescribing

PLOS ONE

Dear Dr. Graham-Clarke,

Thank you for submitting your manuscript to PLOS ONE. After careful consideration, we feel that it has merit but does not fully meet PLOS ONE’s publication criteria as it currently stands. Therefore, we invite you to submit a revised version of the manuscript that addresses the points raised during the review process.

We look forward to receiving your revised manuscript.

Kind regards,

Vijayaprakash Suppiah, PhD

Academic Editor

PLOS ONE

Journal Requirements:

2. Please include additional information regarding the questionnaires used in the study and ensure that you have provided sufficient details that others could replicate the analyses. For instance, if you developed a questionnaire as part of this study and it is not under a copyright more restrictive than CC-BY, please include a copy, in both the original language and English, as Supporting Information.

Reviewers' comments:

Reviewer's Responses to Questions

**Comments to the Author**

1. Is the manuscript technically sound, and do the data support the conclusions?

Reviewer #1: Yes

2. Has the statistical analysis been performed appropriately and rigorously? 

Reviewer #1: Yes

3. Have the authors made all data underlying the findings in their manuscript fully available?

Reviewer #1: Yes

4. Is the manuscript presented in an intelligible fashion and written in standard English?

Reviewer #1: Yes

5. Review Comments to the Author

Reviewer #1: Thank you for the opportunity to review this manuscript exploring NMP in allied health. The manuscript is a useful addition to the literature. The Introduction and Methods are well described and easy to follow.

I only have minor comments that can be readily addressed:

1) Please describe how Kendall's w was interpreted.

2) Table 5 - please amend the p-values to <0.01 or similar. p-values can't be 0.

3) It may be helpful to focus the initial aspects of the Discussion on the results that were obtained. Currently, the whole Discussion reads as the 'strengths and limitations' although I appreciate the thoroughness with which the authors have considered these issues.

6. PLOS authors have the option to publish the peer review history of their article (what does this mean?). If published, this will include your full peer review and any attached files.

Reviewer #1: **Yes: **Brett Vaughan

---

## [Author Response · Author response to Decision Letter 0]

24 Nov 2020

Editor’s Comments 

We have checked the manuscript and amended it where necessary.

Additional supporting information files have been appropriately named

Please include additional information regarding the questionnaires used in the study and ensure that you have provided sufficient details that others could replicate the analyses. For instance, if you developed a questionnaire as part of this study and it is not under a copyright more restrictive than CC-BY, please include a copy, in both the original language and English, as Supporting Information. We have included copies of all the questionaries as supporting information (S2, S3 and S4 Appendices).

The consensus results for Round Three have been added as supporting information to enable comparison with the Round Two results (S4 and S5 Tables)

A worked example of the weighting for ranked results in Round Three has been added as supporting information (S5 Appendix) to clarify the process.

Reviewer’s Comments

Thank you for the opportunity to review this manuscript exploring NMP in allied health. The manuscript is a useful addition to the literature. The Introduction and Methods are well described and easy to follow. 

Thank you for your kind comments and feedback.

Please describe how Kendall's w was interpreted. The criteria for interpreting Kendall’s Coefficient of Concordance (W) are included in Table 1, and we have added an additional sentence to clarify the potential range of results for Kendall’s Coefficient of Concordance (W).

Table 5 - please amend the p-values to <0.01 or similar. p-values can't be 0.

Thank you for highlighting this. We have amended Tables 5, 6 and 7 accordingly and also amended the column heading to further clarify the tables. 

It may be helpful to focus the initial aspects of the Discussion on the results that were obtained. Currently, the whole Discussion reads as the 'strengths and limitations' although I appreciate the thoroughness with which the authors have considered these issues. 

We have reordered the Discussion as you suggest, highlighting the initial findings.

---

## [Decision Letter · Decision Letter 1]

18 Jan 2021

A Delphi study to explore and gain consensus regarding the most important barriers and facilitators affecting physiotherapist and pharmacist non-medical prescribing

PONE-D-20-27040R1

Dear Dr. Graham-Clarke,

We’re pleased to inform you that your manuscript has been judged scientifically suitable for publication and will be formally accepted for publication once it meets all outstanding technical requirements.

Kind regards,

Vijayaprakash Suppiah, PhD

Academic Editor

PLOS ONE

Reviewers' comments:

Reviewer's Responses to Questions

**Comments to the Author**

1. If the authors have adequately addressed your comments raised in a previous round of review and you feel that this manuscript is now acceptable for publication, you may indicate that here to bypass the “Comments to the Author” section, enter your conflict of interest statement in the “Confidential to Editor” section, and submit your "Accept" recommendation.

Reviewer #1: All comments have been addressed

2. Is the manuscript technically sound, and do the data support the conclusions?

Reviewer #1: Yes

3. Has the statistical analysis been performed appropriately and rigorously? 

Reviewer #1: N/A

4. Have the authors made all data underlying the findings in their manuscript fully available?

Reviewer #1: Yes

5. Is the manuscript presented in an intelligible fashion and written in standard English?

Reviewer #1: Yes

6. Review Comments to the Author

Reviewer #1: Thank you for the opportunity to review the revised version of the manuscript. The authors are to be commended for the changes made to the manuscript, particularly the Discussion. The manuscript will be a useful addition to the literature about non-medical prescribers.

7. PLOS authors have the option to publish the peer review history of their article (what does this mean?). If published, this will include your full peer review and any attached files.

Reviewer #1: **Yes: **Brett Vaughan

---

## [Editor Report · Acceptance letter]

22 Jan 2021

PONE-D-20-27040R1 

A Delphi study to explore and gain consensus regarding the most important barriers and facilitators affecting physiotherapist and pharmacist non-medical prescribing 

Dear Dr. Graham-Clarke:

I'm pleased to inform you that your manuscript has been deemed suitable for publication in PLOS ONE. Congratulations! Your manuscript is now with our production department. 

Kind regards, 

on behalf of

Dr. Vijayaprakash Suppiah 

Academic Editor

PLOS ONE